# Development of New Products for Climate Change Resilience in South Africa—The Catastrophe Resilience Bond Introduction

**Thomas Mutsvene * and Heinz Eckart Klingelhöfer ***

Department of Finance and Investment, Tshwane University of Technology, Private Bag X680,
Pretoria 0001, South Africa
* Correspondence: mutsvenet@tut.ac.za or mutsvenethomas@gmail.com (T.M.);
klingelhoeferhe@tut.ac.za (H.E.K.); Tel.: +27-732452185 (T.M.); +27-732310852 (H.E.K.)

**Abstract:** Climate change has brought several natural disasters to South Africa in the form of floods, heat waves, and droughts. Neighbouring countries are also experiencing tropical cyclones, almost on a yearly basis. The insurance sector is faced with an increased level of climate change risk with individuals, corporates, and even the government approaching it for financial cover. However, with an increased level of competition in the insurance sector, (re)insurers must engage in massive product research and development. Therefore, this paper looks at the possibility of the insurance industry developing new products in the form of catastrophe resilience bonds (CAT R Bonds). A qualitative approach is used following content analysis of (re)insurers' product development policies, marketing documents, company reports, and risk management reports as well as the Conference of Parties 27 and 28 resolution papers. The findings reveal that (re)insurers' underwriting capacity, reinsurance protection, and innovative and creative product development increase because of CAT R Bonds. CAT R Bonds enhance the interaction between the capital market and money market, thereby giving speculative investors another investment option. Increased investment into new product development such as CAT R Bonds must continue in South Africa in pursuit of climate change resilience goals.

**Keywords:** climate change resilience; CAT R Bonds; product development; (re)insurance

## 1. Introduction

The past decade has seen the world experiencing a surge in climate change risks and their impact on the economy; South Africa (SA) has not been spared, especially in the past five years (Government of South Africa—GoSA Climate Change Bill 2022; Santam Insurance Barometer—SIB 2023). Floods, heatwaves, droughts, and cyclones are now more common in South Africa (Mutsvene and Klingelhöfer 2022). Significant declines in key sectors such as agriculture, mining, and energy (which negatively affect manufacturing output) have been experienced due to the latent effects of climate change. Several summits such as the Conference of Parties (CoP) and protocols such as the Paris Agreement have been convened and adopted to try and address this challenge (World Bank 2022). The world has accepted a net-zero carbon policy by the year 2050, which is expected to reduce the impact gap of climate change on the global economy in pursuit of United Nations Sustainable Development Goal 13 (UN SDG 13) (United Nations Office for Disaster Risk Reduction—UNDRR 2023).

One dimension in which climate change challenges may be addressed is provided by climate change financing. Governments and the private sector need to focus their attention on the development of new products to achieve climate change resilience. However, the development of such products in the whole of Africa has been relegated to commodity products, and the level of ingenuity is still low (Steve 2019; Van Wyk 2021), with South Africa being no exception (Kirk and Pieterse 2019). Although the catastrophes experienced in the past 5 years show that some parts of South Africa may indeed face a climate change

risk in various economic sectors, no solution has been found to achieve climate change resilience (National Disaster Management Centre—NDMC 2023).

Climate change risks such as drought, flood, wildfire, hail, and weather patterns such as El Niño and La Niña cause tremendous damage to both crops and farm infrastructure in SA (Santam Insurance Barometer—SIB 2023). El Niño is a weather phenomenon that results in less rainfall and higher temperatures across much of Sub-Saharan Africa, while La Niña contributes to periods of above-average annual rainfall in the region. For example, in the 2021/2022 agricultural season, South Africa's hail-related agriculture sector claims reached more than ZAR 2 billion (SIB 2023). However, (re)insurers (insurers and reinsurers) struggle to provide climate change risk coverage not only for the agriculture sector, as other sectors such as energy, mining, and manufacturing are also being severely impacted (Mutsvene and Klingelhöfer 2022). If this situation is left unabated, climate change risks may soon plunge the South African economy into the abyss. Therefore, this paper looks at the possibility of developing CAT R Bonds as financial products to finance climate change resilience in South Africa. The term "resilience" represents both environmental and financial aspects (Więckowska 2013). Climate change resilience is about successfully coping with and managing the impacts of climate change while preventing those impacts from growing worse (Union of Concerned Scientists UCS 2022 and NDMC 2023). The use of financial instruments such as CAT R Bonds may improve climate change financial resilience by providing funding to prevent natural disaster risks impact and developing some initiatives to minimise such impacts.

### 1.1. Research Problem

While climate change risks are on the rise in South Africa, risk financing mechanisms that foster climate change resilience are not yet fully developed (GoSA Climate Change Bill 2022). Partial climate change risk coverage has been traditionally provided by (re)insurance as the magnitude of losses goes beyond the net assets of the (re)insurers (Mutsvene and Klingelhöfer 2022). This has caused (re)insurers to struggle to provide climate change risk coverage, specifically for various sectors like agriculture, energy, mining, and manufacturing. As part of the solution to this challenge, the development of climate change resilience financial products in the form of CAT R Bonds may be tried.

### 1.2. Research Objectives

In aiming to contribute to climate change resilience in SA by developing a CAT R Bond, this paper focuses on the following subobjectives:

- To explain the CAT R Bond product development process.
- To develop climate change resilience products in the form of CAT R Bonds for South Africa.
- To discuss the impact of CAT R Bonds in achieving climate change resilience in South Africa.

## 2. Literature Review

The discourse of environmental finance in SA has gathered momentum in the last decade with various researchers such as Okonjo-Iweala (2017), Steve (2019), Kirk and Pieterse (2019), Van Wyk (2021), Mutsvene and Klingelhöfer (2022), and SIB (2023) having looked into this area. The UN as well as the Government of South Africa have also weighed in through climate change resilience advocacy, policies, protocols, agreements, white papers, and bills. Climate-related financial products are a new type of investment instrument in South Africa (SA) that can work with low-carbon assets and technology to meet the Kyoto Protocol targets (Van Wyk 2021 and Old Mutual Insure 2023). Already, the Kyoto Protocol of 1997, which came into force in 2005, is an international agreement linked to the United Nations Framework Convention on Climate Change (UNFCCC) and aims to reduce carbon dioxide emissions as well as the presence of greenhouse gases in the atmosphere (UNDRR 2023). Although climate change already poses a systemic risk to the

global economy and financial markets (Cevik and Jalles 2022), the green bond market has not been fully exploited in SA (Kirk and Pieterse 2019). However, green standardisations are imperative for the growth of this market (Van Wyk 2021). They can be assisted by effective product development, through a specific process, and for specific people or the market. Product development is a very important process in achieving climate change resilience as it allows for innovation and creativity to develop customised reliant bonds for SA.

### 2.1. Climate Change in South Africa

SA has not been spared by climate change. The years 2017–2022 have seen the country experiencing climate change risks yearly (Mutsvene and Klingelhöfer 2022). Floods, heatwaves, and droughts are now systematic climate change risks in South Africa (SIB 2023). Hence, the SA government developed a National Climate Change Response White Paper in 2012 that translated into policy in 2020 and demands mainstream adaptation in everyday practice and longer-term planning in all spheres and levels of government (GoSA Climate Change Bill 2022). Besides this, climate change research in South Africa has also been growing over the last two decades (SIB 2023). Taking count of South African-authored journal articles, the GoSA White Paper (2012) already highlighted a relatively high number of South African researchers leading and participating in international global-change research programs and scientific bodies, such as the Intergovernmental Panel on Climate Change (IPCC). However, the concentration on climate change resilience finance and building adaptation principles is still in its infancy despite increased experience of climate change risks assumed to stem from high carbon emissions in SA (Steve 2019 and World Bank 2022).

Several sectors of SA have been severely affected by climate change:

- The agriculture sector recorded more than ZAR 2 billion in claims in 2022 alone (SIB 2023).
- Mining processes have been impacted by high carbon emissions and the impact of atmospheric gases on mineral quality (Ziervogel et al. 2014 and Van Wyk 2021).
- The energy sector suffered from climate change's impact on thermal, wind, and photovoltaic energy plants and sources (GoSA Climate Change Bill 2022).
- The country's manufacturing and industrial sectors' raw materials were negatively impacted by high carbon emissions and other climate change variables (Steve 2019).

Therefore, if this crisis triggered by climate change remains unattended, it may cause the South African economy to succumb to climate change risks that usually cause high losses.

### 2.2. Efforts towards Climate Change Resilience

During CoP 27 and CoP 28, South Africa indicated that it had enabled climate-resilient development through integrating global action on adaptation, mitigation, and sustainable development, funding research, promoting the use of indigenous knowledge, supporting data flow, and filling gaps (European Parliament 2022 and UNDRR 2023). Furthermore, it has improved climate literacy and built a capacity for managing complex decision-making (Van Wyk 2021). However, Ziervogel et al. (2014) highlighted the particular need to develop South Africa's capacity to undertake integrated assessments of climate change in support of climate-resilient development planning.

In terms of governance, SA has adopted an inclusive, locally-led, equitable, transboundary cooperation, and benefit-sharing approach (GoSA White Paper 2012). Such an approach makes it easier to manage every climate change risk that impacts a country (Steve 2019). The government has set up appropriate legislation to promote climate change co-ordination within the government (GoSA Climate Change Bill 2022). It has placed its climate change resilience emphasis on ecosystem-based adaptation that focuses on environmental, social, and economic benefits. There have been efforts to address finance gaps and improve finance flows by mobilising billions of US dollars less than adaptation cost

estimates (Van Wyk 2021). However, to date, SA is still grappling with climate change risk financing in various sectors, including agriculture (Mutsvene and Klingelhöfer 2022).

*2.3. Existing Climate Change Resilience Barriers in South Africa*

Despite all efforts being put in by researchers and the increased awareness of climate change risks in South Africa, the country still faces some challenges towards achieving climate change resilience (GoSA White Paper 2012; Mutsvene and Klingelhöfer 2022). In their findings, Steve (2019) and Van Wyk (2021) referred to institutional barriers for addressing climate change that have been identified across a range of South African studies: a lack of capacity (both in terms of numbers of people and expertise), high turnover of staff within government departments; limited understanding of and expertise in tackling climate-related issues; the positioning of climate change as an environmental issue rather than as a development issue; conservative financial management practices; and poor communication and coordination between departments and between different levels of government (especially national to local and provincial to local). Weak relationships between many different stakeholder groups in South Africa (government, civil society, researchers, practitioners, and private sector), may affect climate change adaptation initiatives (GoSA Climate Change Bill 2022; SIB 2023). A climate resilience financing system using CAT R Bonds requires strong interaction between these stakeholders (Vaijhala and Rhodes 2018).

*2.4. Addressing Key Knowledge Gaps in SA's Climate Change Resilience*

Despite nearly two decades of climate and impact research in South Africa, many knowledge gaps in climate change finance resilience remain (Mutsvene and Klingelhöfer 2022). This affects new product development for climate finance resilience, as researchers such as Ziervogel et al. (2014), Okonjo-Iweala (2017), and Steve (2019), specialised in assessing early warning signal detection, socioeconomic vulnerability, and adaptation approaches, have shown. Already, the GoSA White Paper (2012) pointed out that climate change risks in South Africa impact various sectors such as agriculture, mining, energy, manufacturing, and industrials. While, to some extent, some notable climate change risks have been partially covered by (re)insurance, the portion that remains uninsured stretches above ZAR 20 billion in SA (SIB 2023). The reasons for not finding solutions to this widening uninsured climate change risk gap have been attributed to little research and stifled innovation and creativity by insurers:

Global organisations such as the United Nations and World Bank as well as several researchers have pushed for the advancement of climate change resilience research and investments (Więckowska 2013; Vaijhala and Rhodes 2018; Cox 2021; Ando et al. 2022; World Bank 2022; UNDRR 2023). Furthermore, the UN has adopted a net-zero carbon-free world goal by 2050 (CoP 27 and CoP 28) as a way of combatting the effects of climate change, which are believed to have triggered some devastating catastrophe events such as cyclones, hurricanes, earthquakes, tornados, floods, droughts, etc. Also, (re)insurance companies' reports such as Swiss Re (2016), Aon (2017), European Parliament (2022), SIB (2023), and UNFCCC (2023) have indicated that they have focused on ways of reducing or suppressing climate change risks, although UNDRR (2023) notes that increased innovation and product development are still needed. Moreover, while researchers such as Kirk and Pieterse (2019) and Cevik and Jalles (2022) focused on how to manage the economy in times of climate change, more attention is needed on climate change resilience research. Also, this reveals that there has been neglect in the area of climate change resilience finance.

*2.5. CAT R Bond Development Stages*
2.5.1. Overview

Since climate change is a problem that exists in South Africa and her neighbouring regions (SIB 2023), new products need to be developed that deal with climate change resilience. Product development starts with understanding the specifications that the product needs to meet (Cooper 2019). To achieve climate change resilience, developing a

new product in the form of a CAT R Bond may be a solution: The resilience rebate can be used to finance climate change resilience projects (as highlighted under the following bond specification in Section 2.5.2). Furthermore, it provides climate change risk coverage beyond traditional (re)insurance, as it can be used to expand underwriting capacity (Polacek 2018). Investments pooled by the SPV from the CAT R Bond proceeds as well as premiums from sponsors can create funds for improving (re)insurers' climate change risk underwriting capacity (Mutsvene and Klingelhöfer 2022).

According to Tzokas et al. (2003), product development is guided by a new product strategy that aims to align the NPD efforts of the firm with its strategic imperatives. Translating this firm-wide view to country-specific product development, SA must ensure that new climate change resilience products support the strategic objectives of the country and make the best use of its strategic competencies (GoSA Climate Change Bill 2022). In developing the CAT Bond as a new climate change resilience product, Tzokas et al.'s (2003) steps for new product development can be adapted to come up with CAT R Bond development stages as follows:

- Idea generation followed by idea screening to remain with the best idea of climate change resilience instruments to develop;
- CAT R Bond concept development and testing;
- Building a climate change resilience case;
- CAT R Bond development, design, and prototype testing;
- Financial and technical feasibility analysis of the CAT R Bond;
- Market testing and analysing test market results;
- CAT R Bond market launch and post-launch evaluation (short and long term).

Tzokas et al. (2003) primarily focused on the general product development process, with idea generation, concept development, product development, design and prototype testing, market testing, and market launch. The adaptation of Tzokas et al.'s (2003) model involved specification of the product as the CAT R Bond and the transposition of arrows for financial instrument analysis, analysing test market results and short/long term post-launch evaluation. This brought a continuous flow and easy-to-interpret meaning of the CAT R Bond's development stages as other researchers such as Sajid et al. (2015) and Hall (2022) argued that product analysis (financial instrument analysis) precedes market testing, analysis of test market results follows market testing, and all inform short/long term post-launch evaluation, which gives ideas for proper market launch. This is shown in Figure 1 below and equally provides the structure for the rest of this Section:

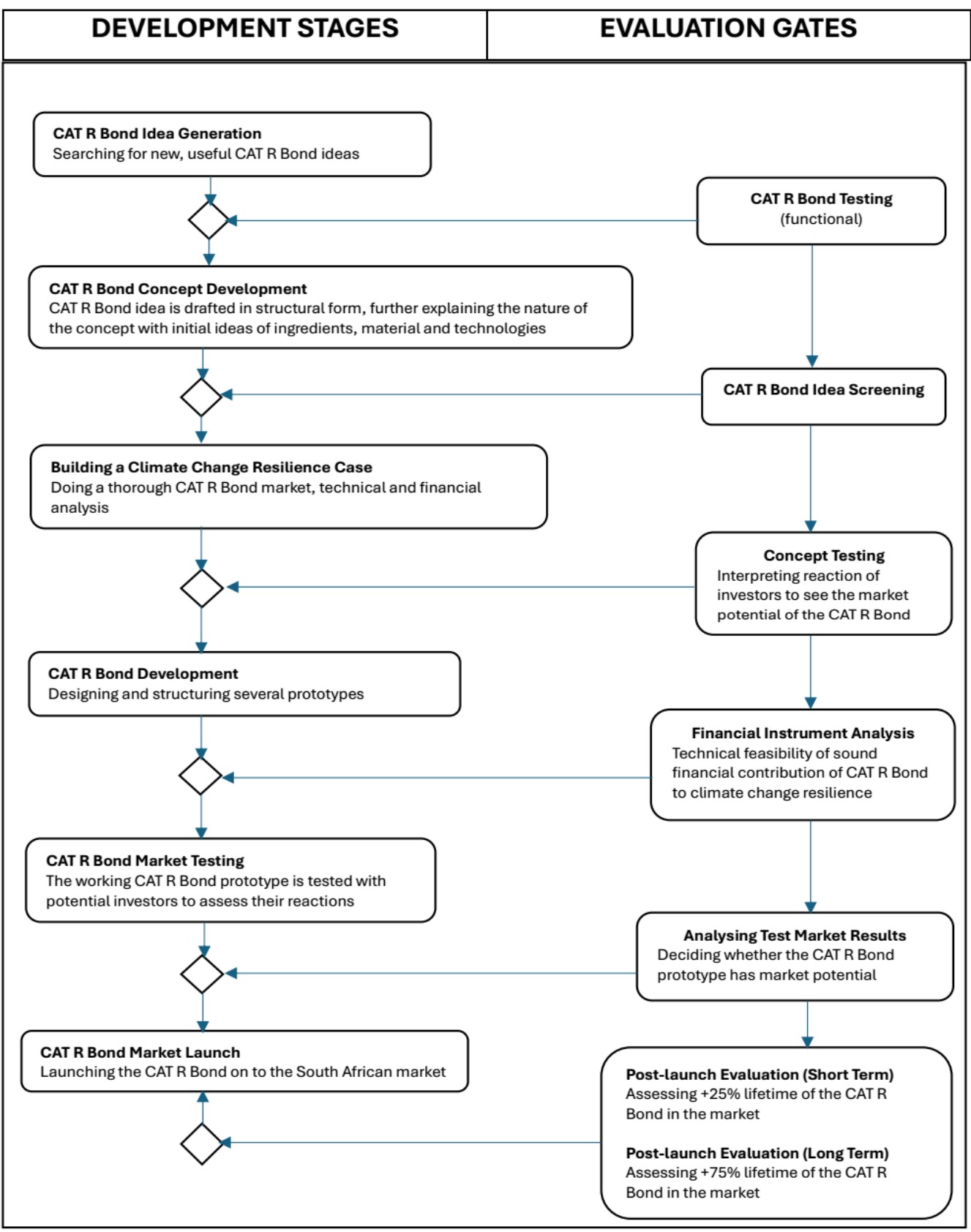

**Figure 1.** CAT R Bond development stages and evaluation gates (Source: Own, adapted from Tzokas et al. (2003)).

### 2.5.2. The CAT R Bond Concept Development

#### CAT R Bond Idea Generation and Climate Change Resilience Funding

In developing the CAT R Bond, idea drafting is carried out to present the bond in structural form, explaining the nature of the concept with initial ideas of ingredients, materials, and technologies to allow for effective idea screening. This helps in coming up with the climate change financial product (the CAT R Bond) that suits a specific climatic environment and provides the (re)insurance industry with a way to provide cover. In South Africa, the development of climate change finance products is very important in ensuring that the gaps of coverage left by (re)insurance can be covered (Van Wyk 2021). Bascunan et al. (2020) posit that even though the CAT R Bond evidence base is nascent, international institutions and national and regional governments are showing an increased appetite for investing in climate adaptation. The development of climate change resilience bonds in the form of CAT R Bonds may help mitigating the rising climate change risks in SA. They were conceptualised in 2015, but only first issued in Europe by the European Bank for Reconstruction and Development (EBRD) in 2019, subscribed for by insurers, commercial banks, and central banks (Dhanjal 2020 and Bascunan et al. 2020). Their main purpose has been for climate change infrastructure development projects in some developed countries in America, Europe, and Asia (Ando et al. 2022). However, CAT R Bonds can also be developed for the South African market by customization of existing resilience bonds to be market- and sector-specific—and the specific climate change risk resilience project.

Bascunan et al. (2020) define resilience bonds as a sub-set of green bonds that seek to raise capital specifically for climate-resilient investment. They further highlight that CAT R Bond investments improve the ability of assets and systems or (re)insurers or governments to persist, adapt, and transform in a timely, efficient, and fair manner that reduces climate risk, avoids maladaptation, and unlocks broader development benefits. CAT R bonds are a form of catastrophe bonds (CAT Bonds) that link insurance premiums to resilience projects in order to monetise avoided losses through a rebate structure (Vaijhala and Rhodes 2018; this will be explained later in this section). The CAT R Bond models two scenarios (Cevik and Jalles 2022): for business as usual and for a protective climate change resilience infrastructure project scenario through the resilience rebate. This allows for estimating the difference in expected losses when the catastrophe happens with and without the project. The rebate can then be used to increase funding for the earmarked climate change risk resilience project (Dhanjal 2020).

Thus, a CAT R Bond is a special type of catastrophe bond for a specific catastrophe or climate change risk, for example, a flood (Cox 2021). Dhanjal (2020) posits that CAT R Bonds provide unique opportunities:

- To hybridise principles in debt securities and insurance policies;
- Ultimately divert available funds into climate-resilient projects that will enhance adaptive capacity.

This predestines them particularly for long-lived infrastructure assets that have to face the test of time and a changing climate (Old Mutual Insure 2023). CAT R Bonds take away the cost of disaster damages and losses from taxpayers, businesses, the insurance industry, and governments. This avoids the treatment of such damages as 'externalities'; hence, they relieve strain in usually strapped capacities to disburse, manage, and coordinate emergency funds (Götze and Gürtler 2022).

#### CAT R Bond Specification—How the Financial Product Works

The CAT R Bond makes pay-outs to cover economic losses when a climate change risk is triggered through (re)insurers who disburse the bond proceeds held by the special purpose vehicle (SPV). This resilience bond is similar to a life insurance policy in the sense that it reduces premiums for the insured's risk-reducing interventions (such as not smoking or regular exercising). Here, it means that premiums for climate-resilient interventions may decrease when the latter allow for reducing the economic losses during their life (e.g., flood barriers) (Dhanjal 2020). Thus, a CAT R Bond can be understood as a merger of a

fixed income security and an insurance policy, as it provides financing for climate change resilience projects and also climate change disaster coverage (Vaijhala and Rhodes 2018). While not necessarily reducing the physical risk(s), this is achieved by the bond's ability to reduce the financial consequences for resilience asset owners (Cevik and Jalles 2022). According to Dhanjal (2020) and Cevik and Jalles (2022), (re)insurers employ CAT R Bond modelers (structuring agents) to design their CAT R Bond and to establish the bond price on two levels:

1. When a triggered event occurs with a climate change resilience project (this means the project reduces the economic losses, and lowers risk exposure and coupon payments);
2. When a triggered event occurs without a climate change resilience project (this means increased economic losses, higher risk exposure, and higher coupon payments).

The difference between the coupon payments for a CAT R Bond covering a climate change risk (1) with a resilience project and (2) without a resilience project (level 1 and level 2 above) is called the 'resilience rebate'. Hence, this resilience rebate can be seen as an incentive to finance climate change risk reduction: while the bond is already covering economic losses from a triggered event (thus also alleviating the burden on the budgets of cash-strapped governments in trying to finance disaster losses (Dhanjal 2020)), the resilience rebate can help to finance a resilience project that in turn will reduce the possible economic losses. This gives reduced investment risk to investors (who can be individuals, or institutional or specialised CAT funds) once the climate change resilience project is complete.

Similar to other bonds, if there is no climate change risk triggered during the lifetime of the CAT R Bond (say 3–10 years), investors recoup their principal plus the accumulated compound coupon. However, different from a 'normal' bond, if the risk is triggered during the bond's lifetime, the actual value of the loss will be covered, so that the investors may lose their investment (principal + compounded coupon) partially or even fully. Therefore, CAT R Bonds may play an important role in both transferring climate change risk from government to the private sector as well as in improving resilience against specific climate perils or liabilities (Vaijhala and Rhodes 2018).

### 2.5.3. CAT R Bond Development

To develop and issue a CAT R Bond, an ecosystem of players ranging from local and state government officials responsible for disaster prevention to (re)insurers, who will pay for the losses, and various sectors that may be at risk is required. With global warming becoming a permanent issue, various stakeholders need to find new and innovative solutions to address the crisis. Capital markets like the Johannesburg Stock exchange (JSE) will be the market players, private partners (individual and institutional) and some specialised CAT funds will be the investors, and public sector entities like SA municipalities, insurers, and reinsurers as well as risk pools will be the sponsors (Kirk and Pieterse 2019). Thus, major insurance players will have to work with the reinsurers, SA municipalities and National Disaster Management Centre (NDMC) in responding to information on disasters, related hazards, and disaster management. Furthermore, it should be noted that the process of developing public interest CAT Resilience Bonds will be slower than issuing a conventional Catastrophe Bond because it is necessary to align the timing of the CAT R Bond issuance not only with the timing of major climate change risks, but also with the respective resilience project to reduce risk and economic losses (Vaijhala and Rhodes 2018).

### 2.5.4. Financial Instrument Analysis—Technical Feasibility

To analyse the (technical) feasibility of the CAT R Bond, (re)insurers work with various stakeholders to provide extra underwriting capacity and resilience for climate change risks (Dhanjal 2020). Given that most (re)insurers fail to provide coverage to these risks (Mutsvene and Klingelhöfer 2022), they test this product together with the capital market (like the JSE) and check if the bond can attract both public and private sector investors' interest (Van Wyk 2021). According to the Coalition for Private Investment in Conservation

(2019), CAT R Bonds face unique market risks relative to conventional CAT Bonds, because of the challenges of not only pricing the near-term changes in risk, but also the long-term benefits associated with the resilience projects against the specific climate change catastrophe risks. The parties involved can provide their observations regarding the covered risk and the related resilience project, so that the modeler (structuring agent) of the CAT R Bond can make all the necessary adjustments with respect to certain parameters like the bond price, tenure, coupon rate, and interval of coupon payment anniversary. Thus, a prototype CAT R Bond can be piloted to solicit the views and perceptions of various stakeholders, especially in areas that may require some adjustments to be made. Where necessary, quantitative modelling beyond qualitative analysis can be carried out, and some tests of the reaction of the bond to climate change can be simulated so that results may be inferred. Various models can be made until one that yields the highest resilience coverage is identified. This is when the CAT R Bond will be set and be ready for test marketing.

### 2.5.5. CAT R Bond Market Testing

In test marketing, CAT R Bond issuers (usually (re)insurers) will assess the response of the bond market by testing the potential response of investors to the bond. This is performed by having the CAT R Bond to be tested pre-launched into a particular market (Cooper 2019). This market can be a municipality such as the Tshwane Municipality or the flooding-exposed South Coast in SA. A prospectus may be published inviting investors to express interest in investing in the bond (Edesess 2015). Due diligence (in the form of careful analysis and screening) can be performed on the CAT R Bond investors by the SPV to avoid obtaining funds from unscrupulous investors (Götze and Gürtler 2022). The statistics of investors and their investment amount into the collateral account pool then allows for estimating the total CAT R Bond pooled funds for possible climate change risk coverage as well as for funding climate change resilience projects (Dhanjal 2020).

Alternatively, market testing may be conducted by simply having panel discussions and forums between the possible issuers and interested parties or organisations (Tzokas et al. 2003). These can be (re)insurers being supported by SA's National Disaster Management Centre (NDMC) and various municipalities' disaster management departments hosting such discussions and forums. The participants can then provide instant feedback on any areas that may need restructuring, remodelling, or mere adjustments to make the bond appealing to potential investors and to match the climate change resilience expectations for SA. Once the potential to perform well in the market is established, a market launch for the CAT R Bonds can be held.

### 2.5.6. CAT R Bonds Market Launch

With the continuous surge in climate change risks in SA, it may be useful to establish a catastrophe risk exchange market where CAT R and other related bonds can be traded. Such a market can work together, e.g., with the Johannesburg Stock Exchange (JSE) and other capital markets, to raise funds for climate change or natural disaster financing (Okonjo-Iweala 2017 and Mutsvene and Klingelhöfer 2022). The CAT R Bond will be issued by underwriters who publish the prospectus for speculative investors to subscribe through payment of an application fee as well as an investment amount for a specified number of bonds (Cevik and Jalles 2022). The issuers then undertake investor vetting and screening so that the bonds can be sufficiently allotted to interested applicants (individual, corporate, private, or public investors), generating high enough lumpsums for climate change financing (Bascunan et al. 2020). Obviously, this process requires effective bond selling and marketing skills. For the effective marketing and selling of CAT R Bonds, the catastrophe risk exchange market may even be virtual and operating 24/7 (Morana and Sbrana 2019). This allows for CAT R Bonds not only to attract local, but also foreign investors, thus letting the market become global. This process could be supported by tax incentives. However, going virtual and global makes thorough investor vetting and screening even more of a necessity to counteract the accompanying opportunity for unscrupulous investors to wash

their dirty money (Vaijhala and Rhodes 2018). Also, it is important to perform a post-launch evaluation that focuses on both the short term (25% bond lifetime) and long term (75% lifetime of the CAT R Bond) in the market. This helps in seeing the performance of the bond as a climate change resilience financial instrument and allows for the identification of ways to improve the bond's structure so that it yields results that best benefit both investors and sponsors.

*2.6. Variables from the Literature Review Needed for the Design of the CAT R Bond*

Based on the literature review above, some variables can be extracted for the design of a CAT R Bond. These variables include the following:

- Variable 1—*Rebate* refers to the difference between the coupon payments for a CAT R Bond covering climate change risk with a resilience project and without a resilience project (Dhanjal 2020). It an incentive to promote climate change risk reduction by covering economic losses from a triggered event and alleviating the burden on the budgets of cash-strapped governments in trying to finance disaster losses.
- Variable 2—*Sponsors* provide the necessary CAT R Bond finance through premiums that increase the special purpose vehicle (SPV) pool, usually in exchange for contingent payment in times of climate change claims. They can include (re)insurers, SA municipalities, risk pools, businesses, or corporates.
- Variable 3—*Investors* provide a principal as an investment into the CAT R Bond issued and managed by the SPV. These investors can be in the form of individual or corporate reinsurance financiers are willing to assume speculative risk in a CAT R Bond transaction. Investors receive coupons as a return on their investment. If a climate change risk is triggered, the investors may partially or completely lose their principal depending on the magnitude of loss. Such investors participating in the bond proceeds include individuals, institutions, and some specialised CAT funds.
- Variable 4—The *issuer/special purpose vehicle (SPV)* is responsible for keeping or investing (at nominal value) the CAT R Bond pooled funds in a collateral account to generate short-term investment returns, which may increase the pooled funds or offset any variability in SA Rand (ZAR) value caused by inflation.
- Variable 5—The funds from CAT R Bond investors are kept in a *collateral account* in escrow for a stipulated time. They are to be released to cover catastrophe losses or (if not fully claimed) to pay back the investors' principal and coupons.
- Variable 6—The *risk modeler* is responsible for climate change resilience modelling.
- Variable 7—*Climate change-resilient projects* are aimed at funding climate change risk reduction by mitigating possible damages and reducing expected losses.

## 3. Materials and Methods

This research followed a qualitative documentary and content analysis approach. Publicly available data were used; hence, no ethical risks were foreseen. For the selected journal articles, keywords such as 'climate change resilience', 'Catastrophe Resilience Bonds', 'CAT R Bonds', 'catastrophe bonds', 'product development', and '(re)insurance' were employed. Further screening of titles, abstracts, and publication years resulted in thirty-four (34) articles for the review process. Insurers' and reinsurers' product research and development policies, marketing strategy documents, and company reports for the period 2020–2023, as well as existing regulatory bodies reports and legal frameworks in South Africa, were reviewed to assess coverage gaps calling for new product development like a catastrophe bond. The United Nations (UN) climate change goals and conference resolutions such as the Conference of Parties (CoP) 27 and CoP 28 were studied, including the UNDRR, European Parliament, and UNFCC initiatives, to see how climate change resilience can be improved, possibly by the development of CAT R Bonds, towards the net-zero 2050 goal. In doing so, existing product development processes and models were used in the design of a new climate change resilience financial product in the form of a CAT R Bond. This was achieved by customising various resilience bonds that have been used in

developed cities like Houston, San Francisco, and Miami in North America to design the CAT R Bond, which is a climate change resilience-specific bond.

Since the research used publicly available data and information, no research ethics clearance was required.

## 4. Results (Designing a CAT R Bond for SA)

*CAT R Bond Structure Development*

In this section, a CAT R Bond structure is developed. As explained in Section 2.5.3, to structure a CAT R Bond to provide coverage for climate change risks, various stakeholders such as sponsors, investors, SPVs, and government departments implementing climate change resilience projects like the NDMC work together. It shall be based on the variables extracted from the literature review as listed under Section 2.6.

As a main component of the CAT R Bond structure, the SPV maintains a safe collateral account while investing part of the principal in risk-free to near-risk-free money market instruments. Thus, already the CAT R Bond structure helps to reduce insurance premiums to sponsors (insurers/reinsurers) as well as the risk to investors' principal. Insurers can use funds generated from a CAT R Bond transaction to finance coinsurance for climate change risks or raise their retention capacity. Reinsurers can stretch their underwriting capacity to accept ceded climate change risks, thus raising their book of business as well as profitability. Due to the good efforts towards climate resilience projects, the sponsors (and via them also the insurance premiums for the covered risks) can be rewarded by reduced premiums as a way of encouraging the resilience culture – just as what occurs regarding good health habits in life and health underwriting. Furthermore, the resulting reduction in possible losses with respect to the covered risks also contribute to make a CAT R Bond more attractive than a "normal" Catastrophe Bond to the investors. Hence, such win–win advantages to both sponsors as well as investors help attract their participation in the CAT R Bond transaction.

Taking all this into account, in the end, the SPV has a double function:

1. It creates and issues the CAT R bond, taking into account the results from the risk modelling process as just described and creating the win-win situation for both the investors as well as the sponsors.
2. It receives the (reduced) premiums from the sponsors and the bond proceeds from individual/corporate investors holds these pooled funds in the collateral account and invests them into products at the money market investments that can be easily liquidated at short notice. The returns can be used to make contingent payments to sponsors as well as to repay investors' principal and coupon.

By joining all the extracted variables, Vaijhala and Rhodes's 2018 infrastructure resilience model can be further developed into a CAT R Bond structure as shown in Figure 2. As a result of this, the underwriting capacity and/or the level of preparedness to cover climate change risks of the sponsors gets improved, while risk-taking investors get a product that promises less risks (thus a higher security equivalent) for their repayments, making the investment more attractive than a "normal" Catastrophe Bond:

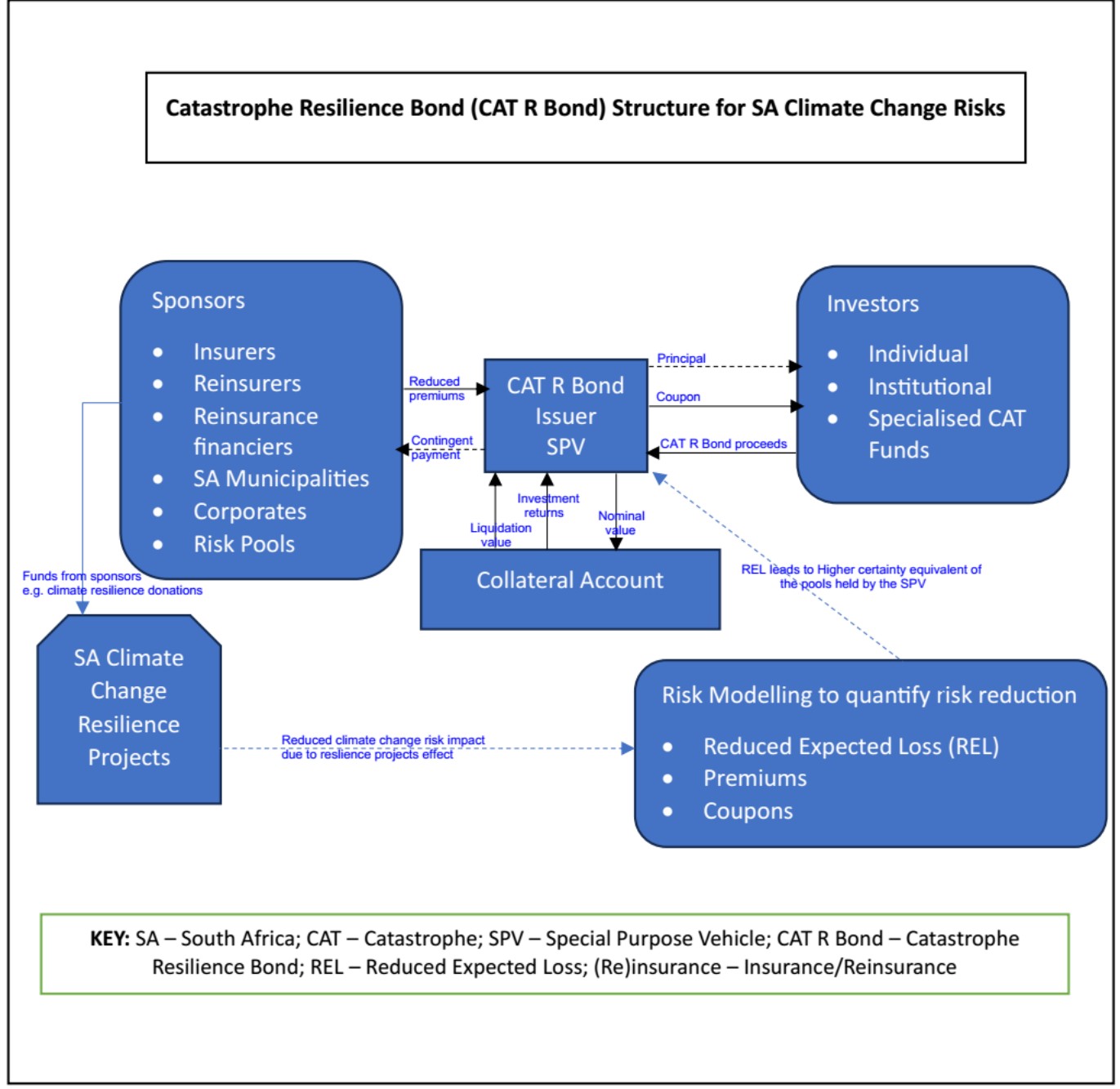

**Figure 2.** CAT R Bond structure for SA climate change risk (Source: Own, adapted from Vaijhala and Rhodes 2018).

## 5. Discussion

### 5.1. Explanation of the Designed CAT R Bond Product

The designed CAT R Bond for SA helps reducing expected losses from specified natural disasters such as floods, wind, hail, heatwaves, and drought. This type of bond

- like "normal" Catastrophe Bonds provides funding for climate change risk reduction by covering economic losses from a triggered event, thus, alleviating the burden on the budgets of cash-strapped governments in trying to finance disaster losses (Dhanjal 2020),
- incentivises additional resilience projects through the resilience rebate (thus, extending possible risk cover even further).

This also helps in improving SA's risk reduction potential in pursuit of the global call to build climate change-resilient economies through global stock take, climate and health management, loss and damage analysis, climate finance, and global action on adaptation as advocated by UN SDG 13, CoP 27, CoP 28, and the UNDRR's agenda (European Parliament 2022 and UNFCCC 2023).

The designed CAT R Bond (as a financial instrument product) engages interested resilience bond sponsors such as reinsurance financiers, (re)insurers, municipalities' disaster management departments, private businesses/corporates, and risk pools. They address the SPV (special purpose vehicle), which acts as an interface to the investors (Polacek 2018). To make an attractive and sustainable offer to both sponsors and investors, the SPV works with CAT risk modelers who carry out risk quantification by characterising the specific climate change risks to be covered, determining the bond price and the coupons. They also estimate the reduced expected loss (REL) following the realisation of the resilience projects financed by the rebate while the rest goes for premiums and bond repayments (including coupons). The REL represents the reduction in climate change-triggered expected CAT losses that result from the implementation of the funded climate change resilience projects (for example, the reduction in expected harvest losses after the construction of high wind-breaking walls or after installing hail-resistant greenhouses).

Following the bond's advertisement, the SPV carries out thorough vetting and screening of possible investors before accepting their CAT R Bond proceeds amount. Since the SPV will be regulated by a legal framework and controlled by existing financial regulators in SA, the paid funds and premium contributions kept in the collateral account will be safe for investors (Mutsvene and Klingelhöfer 2022). The pooled funds can be used for either repaying the investors (if a covered climate change catastrophe is not triggered) or covering the claims (if a catastrophe is triggered):

- The investors invest speculatively in the bond with the hope of receiving a return in the form of a higher coupon than normal if the climate change catastrophe is not triggered, but they may lose the principal and coupons partly or even fully should a specified climate change risk be triggered.
- The CAT R Bond stimulates climate change resilience action in specified communities, which results in (re)insurance rebates on premiums. Such climate change resilience action reduces the risk to the investors' principals, which, in turn, increases the chance of receiving a (higher) return on their investment (in the form of a coupon).
- The premiums charged by sponsors (such as (re)insurers) will also be reduced because of the rebate given in appreciation of the climate change resilience projects proactively carried out.

Thus, the CAT R Bond provides extra climate change coverage beyond traditional (re)insurance coverage. This is further improved by the extra benefit that comes to both the sponsors and investors in the form of reduced premiums and reduced risk to the investment principal, respectively, as a result from the implementation of the specific climate resilience projects (Więckowska 2013).

*5.2. Impact of the CAT R Bonds on Climate Change Resilience in SA*

Resilience bonds have proven useful in the Western world (Polacek 2018) and can be replicated as CAT R Bonds in developing economies like SA to improve catastrophe resilience. Through modelers'/structuring agents' assistance in CAT R Bond structuring and customisation, the bonds can expand financial protection for SA communities vulnerable to climate change. Through proactive funding by offering rebates and reactive disaster recovery actions, they help to leverage new climate finance resilience that offers a measurable risk impact reduction. Hence, these bonds can bring innovation for SA to initiate climate change projects with resilience in mind.

Furthermore, bringing in (private) investors can also improve the (re)insurance industry's underwriting capacity (Old Mutual Insure 2023 and NDMC 2023). Considering that climate change risks such as heatwaves, droughts, floods, wildfires, hail, and weather

patterns such as El Niño and La Niña are becoming more common in SA and have multi-sectorial impact, CAT R Bonds may bring a financing option to reduce expected losses. Once this coverage is guaranteed, the confidence of businesses in various sectors will, in theory, be boosted, thereby motivating them to produce more, which spurs economic growth.

*5.3. Practical and Managerial Implications*

Several practical and managerial implications can be derived from this research. The following are some of these implications:

- The CAT R Bond improves SA's climate change resilience strategies as it provides an option to cover climate change risks beyond traditional (re)insurance.
- As an alternative to traditional (re)insurance, the CAT R Bond enhances the capacity of insurers through higher retention or increased coinsurance participation ability and reinsurers through improved underwriting capacity for climate change risks.
- It brings the concept of co-sponsoring climate change resilience initiatives between the insurance sector, private sector, and government municipalities as well as the National Disaster Management Centre (NDMC).
- Since CAT R risk modelers must deal extensively with anticipated risks and losses of climate change, quasi as a byproduct, they can create models that improve practical disaster preparedness and post-disaster recovery processes rather than just the traditional vulnerability and impact assessments.
- The CAT R Bond is a variation of catastrophe bonds that does not only cover possible losses resulting from climate change catastrophes, but that can also leverage humanitarian aid and international development funding for disaster risk reduction projects around SA since—via the premium rebates—they support, in particular, the funding of climate change resilience projects. This is in line with the advocacy of the United Nations Office for Disaster Risk Reduction (UNDRR).
- The rebate from (re)insurance savings may serve as extra funds that can be directed towards financing climate change resilience projects.

*5.4. Recommendations*

The adoption of the CAT R Bond as part of climate change finance resilience techniques in SA is highly recommended given its ability to promote the possibility of the insurance sector to underwrite climate change risk. While this may be a working model to increase cover against climate change risks as well as to finance climate change resilience projects in SA, further quantitative tests and modelling of the CAT R Bond may be needed to quantify the magnitude of the (re)insurance coverage leverage given to the insurance industry. (Re)insurers are advised to invest in innovation that capacitates them to underwrite climate change risks as indicated by the Swiss Re (2016), Aon (2017), and SIB (2023) reports. The first entrants into providing climate change CAT coverage might enter a lucrative business niche, as past research by Steve (2019), Kirk and Pieterse (2019), Cevik and Jalles (2022), and Van Wyk (2021) has established that most (re)insurers avoid underwriting climate change risks. It is also recommended that the (re)insurance industry collaborates with the private sector and government departments such as the NDMC and municipalities in combatting climate change risks, as lack of such collaboration stifles the efforts of product development and providing climate change risk coverage. Continuous product development in response to the global call for action against climate change such as the UN net-zero goals of decarbonisation, Kyoto protocol for reducing greenhouse gas emissions, and CoP resolutions for combating climate change can play a significant role in achieving climate change resilience in SA. In a similar way, also other countries may use the CAT R Bond concept for climate change risk financing through customisation to suit country-specific metrics.

## 6. Conclusions

CAT R Bonds have been designed with the conviction that proactive planning of catastrophes is more cost-effective than post-disaster reconstruction. These bonds are designed for the following:

- On the one hand, to provide cover against the losses resulting from catastrophic events;
- On the other hand, to monetise losses avoided by the realisation of climate change resilience projects.

Hence, they help bring in investors to invest proactively in proactive risk reduction climate change projects. The potential for South African municipalities to finance climate change resilience projects, share the burden with other stakeholders like (re)insurers, and transfer the risk of a catastrophe to speculative (private or institutional) investors via capital markets can improve the country's climate change resilience. Thus, these bonds allow for providing climate change cover beyond traditional (re)insurance, thereby helping in building a climate change-resilient economy in South Africa.

**Author Contributions:** Both authors contributed equally to this paper. This includes conceptualisation, methodology, validation, investigation, writing, review, editing, and visualisation. All authors have read and agreed to the published version of the manuscript.

**Funding:** This research received no external funding. Neither author received any funding beyond their academic appointment.

**Data Availability Statement:** No new data were created or analyzed in this study. Data sharing is not applicable to this article.

**Conflicts of Interest:** There was no conflict of interest in the write-up of this paper. Both authors declare that they have no financial or personal relationships that may have inappropriately influenced them in writing this article.

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
