# Peer review of "Development of New Products for Climate Change Resilience in South Africa—The Catastrophe Resilience Bond Introduction"

_jrfm, doi:10.3390/jrfm17050199_

Round 1
Reviewer 1 Report
Comments and Suggestions for Authors
This study seems to be more like a review paper, and the number of references is small, which is difficult to support the theoretical basis of research progress and research background.
It is recommended to increase the number of references.
It is suggested to organize the research background in the form of a timeline, so that the research background of the research will be more obvious.
Do the conclusions of this study have certain universality and spatial portability in other countries? It is suggested to add relevant content to the discussion.
Comments on the Quality of English LanguageMinor editing of English language required
Author Response
Dear Reviewer 1,
Kindly find the table of corrections attached with actions done to every comment made during the review of the paper.

Reviewer 2 Report
Comments and Suggestions for Authors
Review Report (April 25, 2024)
Development of New Products for Climate Change Resilience in South Africa – The Catastrophe Resilience Bonds Introduction (JRFM 2988520)
Summary:
Climate change in South Africa and neighboring countries has intensified natural disasters like floods, heatwaves, and cyclones, increasing risks for insurers. To address this, insurers are exploring catastrophe resilience bonds (CAT R Bonds). This study analyzes insurers’ strategies and market dynamics, finding that CAT R Bonds enhance resilience and attract investment. Such innovations are crucial for climate adaptation in South Africa.
As both a reviewer and reader, I appreciate sharing research contributions. However, I have several suggestions for enhancing the paper’s quality.
Suggestions:
[1] “A qualitative approach is used following content analysis of (re)insurers’ product development policies, marketing documents, company reports, and risk management reports as well as the Conference of Parties 27 and 28 resolution papers.” (lines 13-27) This quote should be both precise and impactful as it pertains to the research methods outlined in the paper.
[2] The paper’s title is “Development of New Products for Climate Change Resilience in South Africa – Introducing Catastrophe Resilience Bonds.” The term “resilience” is used twice, representing both environmental and financial aspects. However, the concept of resilience is not adequately defined, especially in the Introduction section.
[3] The authors frequently repeat “catastrophe resilience bonds (CAT R Bonds).” To streamline, I recommend using “CAT R Bonds” after the initial mention, retaining the full term only once.
[4] Are CAT R Bonds the only “New Products” discussed in the paper? The findings presented may have broader implications for other “New Products” introduced by insurers and reinsurers. However, insurers and reinsurers serve distinct roles in intermediary functions. Furthermore, can reinsurance financiers engage in transactions involving CAT R Bonds?
[5] Section 2 is dedicated to Materials and Methods, while Section 3 pertains to Literature Review. The arrangement of these sections appears unconventional. Perhaps Section 2 should be swapped with Section 3 to align the structure more logically.
[6] Section 2 focuses on Materials and Methods; however, it lacks references. Is this section intended to be the primary contribution to the literature?
[7] The paper appears two crucial adapted figures: Figure 1: Catastrophe Resilience Bond (CAT R Bond) Development Stages and Evaluation Gates (Source: Own, adapted from Tzokas et al. (2003) and Figure 2: Catastrophe Resilience bond (CAT R Bond) Structure for SA Climate Change 483 risks (Source: Own, adapted from Vaijhala & Rhodes, 2018). However, the authors fail to provide detailed explanations of their adaptations, which diverge from those of Tzokas et al. (2003) and Vaijhala and Rhodes (2018). The purposes behind these adaptations remain unclear. More critically, while both figures address CAT R Bond, Climate Change Resilience is only mentioned in passing.
Author Response
Dear Reviewer 2,
Please kindly find the attached table of corrections with actions done for every comment made during the review of the paper.

Round 2
Reviewer 1 Report
Comments and Suggestions for Authors
Accept in present form
Reviewer 2 Report
Comments and Suggestions for Authors
The authors have responded adeptly to all suggestions.
The authors have responded adeptly to all suggestions.
The authors have responded adeptly to all suggestions.
The authors have responded adeptly to all suggestions.
The authors have responded adeptly to all suggestions.
The authors have responded adeptly to all suggestions.
The authors have responded adeptly to all suggestions.
The authors have responded adeptly to all suggestions.
The authors have responded adeptly to all suggestions.
The authors have responded adeptly to all suggestions.
The authors have responded adeptly to all suggestions.
The authors have responded adeptly to all suggestions.
The authors have responded adeptly to all suggestions.